# Mucosal leishmaniasis is associated with the *Leishmania* RNA virus and inappropriate cutaneous leishmaniasis treatment

Fredy A. Pazmiño[1], Marcela Parra-Muñoz[1], Carlos H. Saavedra[2], Sandra Muvdi-Arenas[3], Clemencia Ovalle-Bracho[4], María C. Echeverry[1]*

1 Departamento de Salud Pública, Facultad de Medicina, Universidad Nacional de Colombia, Bogotá, Colombia, 2 Departamento de Medicina, Facultad de Medicina, Universidad Nacional de Colombia, Bogotá, Colombia, 3 Hospital Universitario Centro Dermatológico Federico Lleras Acosta, Bogotá, Colombia, 4 Facultad de Ciencias, Pontificia Universidad Javeriana, Bogotá, Colombia

* mcecheverryg@unal.edu.co

**Data Availability Statement:** All relevant data are within the manuscript and its Supporting Information files.

## Abstract

### Background

Mucosal leishmaniasis (ML) is a severe clinical form of leishmaniasis that is characterized by the destruction of the nasal and/or the oral mucosae and appears as a late complication in 5% to 10% of cutaneous leishmaniasis (CL) cases produced by species belonging to *Leishmania* (*Viannia*) subgenus. Some strains of *Leishmania* spp. carry an RNA virus known as *Leishmania* RNA virus (LRV) that may contribute to the appearance of ML.

### Methods

To examine the role of LRV type 1 (LRV1) as a risk factor associated with ML, a retrospective case-control study involving 103 patients was conducted. Cases were defined as patients with ML (n = 33), and controls corresponded to patients with CL and without mucosal lesions (n = 70). Clinical data were recorded from the patient's medical records. Cryopreserved biopsies were used to detect LRV1 and identify *Leishmania* species.

### Results

The frequency of LRV1 in the 103 patients was 16.5% (95% CI,10.4–25.12) being higher in samples from cases [33.33% (95% CI,18.89–51.76) than from controls [8.57% (95% CI, 3.82–18.10)]. *L. (V.) braziliensis* was identified in 63.6% of cases and 55.7% of the controls. Multivariate logistic regression indicated that infection with *Leishmania* spp. carrying LRV1 (OR = 6.30; 95% CI,1.52–26.10, p = 0.011) acts as risk factors for ML occurrence, while the completed treatment for the cutaneous event decreases the risk of ML (OR = 0.039; 95% CI, 0.01–0.12, p < 0.0001).

### Conclusions

Our data support the association between LRV1 and ML occurrence and emphasize the effect of completed treatment for CL in preventing ML.

**Funding:** ME Grant code 110177758491. This work was supported by the Administrative Department of Science, Technology, and Innovation (Colciencias) of Colombia The funders had no role in study design, data collection and analysis, decision to publish, or preparation of the manuscript.

**Competing interests:** The authors have declared that no competing interests exist

## Introduction

Leishmaniasis is a disease caused by a group of protozoan parasites belonging to the genus *Leishmania*, transmitted to mammals via the bite of infected Phlebotomine sandflies. In humans, leishmaniasis has three major clinical manifestations: cutaneous leishmaniasis (CL), mucosal leishmaniasis (ML), and visceral leishmaniasis. It is distributed widely in tropical and subtropical regions, affecting 98 countries [1]. ML corresponds to lesions with significant tissue destruction occurring mainly on the oropharyngeal and/or nasal mucosa, that appears as a systemic complication in a proportion of patients either months or years—usually one to five years—after cutaneous lesions have healed [2], predominantly in the South American region [3]. ML seriously affects patients' quality of life due to treatment failure, relapse, and slight or severe facial deformity [2, 4].

*Leishmania* species belonging to the *Leishmania* (*Viannia*) subgenus, such as *L.* (*V.*) *braziliensis*, *L.* (*V.*) *guyanensis*, and *L.* (*V.*) *panamensis*, are predominant in South America and associated with the occurrence of ML. In CL cases produced by *L.* (*V.*) *braziliensis*, pentavalent antimonial treatment failure is higher when compared with cases from other members of the *L.* (*V.*) subgenus [3, 5–7]. In addition to the parasite species, studies have found host factors such as the patient's age and sex and immune response gene polymorphism to be involved in the appearance of ML [8–14]. It is widely accepted that the host's immune response is of great relevance in the pathophysiology of ML [15, 16].

LRV is a cytoplasmatic double-strand RNA virus associated with some *Leishmania* spp. strains [17, 18]. Experiments involving mice have shown that the infection with *L.* (*V.*) *guyanensis* carrying *Leishmania* RNA virus type 1 (LRV1) promotes inflammation and subverts the host immune response [19]. Mice infected with *L.* (*V.*) *guyanensis* LRV1–negative but that were simultaneously infected with exogenous dsRNA viruses, such as lymphocytic choriomeningitis virus (LCMV), showed immunological and clinical profiles equivalent to those presented by isogenic mice infected with *L.* (*V.*) *guyanensis* LRV1–positive [20]. Comparison of *in vitro* infections of human macrophages by LRV1–negative with infections by LRV–positive strains of *L.* (*V.*) *braziliensis* revealed that the cytokines' expression profile in the latter case showed Th2 predominance [21].

Therefore, it is clear that the presence of LRV1 does not go unnoticed by the host immune system. However, the clinical relevance of human infections with LRV1-carrying *Leishmania* strains is unknown. Two studies consistently suggest that LRV1 is associated with therapeutic failure in CL [22, 23], but findings regarding the role of LRV1 in the occurrence of ML are inconclusive [24–27].

The present work corresponds with a retrospective case-control study involving 103 Colombian patients suffering from either ML or CL to assess the association between ML and LRV1. The main findings from this study support this association but additionally emphasize the protective role of a complete treatment for the cutaneous leishmaniasis event in the occurrence of ML.

## Methods

### Ethical approval

The Ethical boards of the participating institutions granted ethical approval for this study. The study followed national (Resolution 008430 of the Ministry of Health of Colombia) and

international (Declaration of Helsinki and amendments, World Medical Association, Korea 2008) regulations. All clinical samples had been taken from patients as part of routine diagnosis and treatment, with no unnecessary invasive procedures, and with written informed consent. This consent authorizes the use of the stored sample, analyzing clinical records, and updating the contact details. A new informed consent was signed for the 19 patients who could be localized and were involved in the clinical examination. For underage patients involved in the study, informed consent was obtained from parents and/or legal guardians.

## Sample size, patients, and samples

This was an unpaired retrospective case-control study with a two-sided confidence level of 95%, power of 90%, a case: control ratio of 1:2, and estimated frequencies of exposure to LRV1 between cases and controls of 71.1% and 36.7%, respectively [24]. Based on these parameters, the minimum sample size required was 33 cases and 66 controls using the Fleiss formula.

Recruitment was carried out from May 13, 2019, to February 13, 2020, by reviewing the clinical records of a cohort of 323 patients who attended the leishmaniasis clinic at the Hospital Universitario Centro Dermatológico Federico Lleras Acosta (CDFLLA) from January 1997 to December 2017 and whose cryopreserved biopsies were available at the CDFLLA biobank.

Cases were defined as patients diagnosed with ML who were treated in CDFLLA between 2007 and 2017 and met the criteria described by Ovalle-Bracho et al. [28]. Briefly, Clinical suspicion of ML, histopathological findings suggestive or conclusive of ML, clinical response to treatment (Defined as the improvement of signs and symptoms one month after starting treatment with pentavalent antimonials), Montenegro test, scar suggestive of CL, and epidemiological antecedent. The clinical samples (cryopreserved biopsies) were examined at the Parasitology Laboratory, Faculty of Medicine of Universidad Nacional de Colombia, to establish the presence of *Leishmania* spp., via qPCR for the 18S ribosomal RNA gene the primers referred by van den Bogaart E [29] were used following the protocol modified as described [30]. Biopsies with qPCR 18S-negative for *Leishmania* spp. were excluded. Subsequently, 33 patients composed the case group (S1 Fig).

Controls correspond to patients diagnosed with CL who were treated at CDFLLA between 1997 and 2010. During that period, CL was defined as a "Patient with skin lesions who lives or has visited an endemic region and who meets at least three of the following criteria: evolution greater than two weeks, round or oval ulcer, ulcer with raised edges, nodular lesions, plaque lesions, satellite lesions, and localized adenopathy. Plus, the patient must have a parasitological and /or histopathological test where the *Leishmania* parasite was detected" [31]. The control group was composed of 70 CL patients. For this group, 32 patients' clinical records mentioned a follow-up free of ML symptoms up to 6 months, 14 patients were followed up from 6 to 48 months, and records referred them as free of the ML symptoms; the research team from the current study examined 19 patients between September 7, 2019, and February 29, 2020, assessing all of them as ML-free. In this last sub-group, the median of the time elapsed between the end of the treatment for CL and the clinical examination was 16.83 years (95% CI, 14.49–20.0). Five patients from the total control group did not have follow-up after the treatment for the CL event.

## Clinical records review and variables definition

A manual review of the clinical records of cases and controls was conducted to obtain clinical, epidemiological, and demographic data. The dependent variable was the development of ML, and the independent variable was the presence of LRV1 in the clinical sample.

Other independent variables considered of interest for the analysis and may act as confounding factors were patient gender and parasite species identified in the biopsy. Variables regarding the CL event were analyzed, such as the age when the CL diagnosis was made, geographical origin, anatomical location of the skin lesion, number of skin lesions, and variables associated with the treatment for the CL event. Complete treatment for CL was considered when patients received at least one full round of any of the following regimes independently from the therapeutic outcome: antimony salts (20 mg/kg/ day) for 20 days or miltefosine (1.5 to 2.5 mg/kg/ day) for 28 days, or ketoconazole (600 mg/day) for 28 days.

Variables with missing data higher than 20% were excluded from the analyses. All the information on the variables was recorded and coded in an Excel database for subsequent analysis.

### Clinical samples procedures

Dry frozen biopsies maintained at −80˚C (specimens corresponding with cutaneous tissue for controls and mucous tissue for cases) were processed to confirm infection by *Leishmania* spp. and LRV1 detection. RNA and DNA were extracted from biopsies using the Quick-DNA/RNA Miniprep™ Plus Kit (Zymo Research); 100 ng of RNA quantified by nanodrop was used for the cDNA synthesis using a High-Capacity cDNA Reverse Transcription Kit (Thermo Fisher Scientific) according to the manufacturer's protocol—*Leishmania* spp. Infection was confirmed by either RT-qPCR or qPCR of the 18S ribosomal subunit gene of the parasite [30]; details on *Leishmania* spp. detection and classification are presented in S3 Table. cDNA quality was assessed via RT-qPCR of a human gene, encoding the ribosomal protein L27 (RPL27) [32] in order to avoid false negatives associated with long-term storage samples, following the algorithm presented in S2 Fig. LRV1 detection was performed with modification as the protocol previously described [30]; the reactions were performed in 20 μl final volume, using SYBR Select Master Mix (Applied Biosystems), containing 400 nM of primers designed by Pereira L de O [27] on an Applied Biosystems 7500 Real-Time PCR System and CFX96™ Real-Time PCR Detection System (Bio-Rad). The cycle threshold (Ct) values and melting temperature (Tm) were determined using 7500 Software v2.0.6 and Bio-Rad CFX Manager 3.1, version 3.1. Infecting parasite species in the biopsies were identified via PCR amplification of the *miniexon* gene [33] and PCR-RFLP of the *hsp70* gene [34], using as controls MHOM/BR/75/M4147, MHOM/PA/71/LS94, MHOM/BR/75/M2903, MHOM/BZ/82/BEL 21 and MHOM/BR/73/M2269.

### Statistical analysis

Statistical analysis was carried out in Stata ® version 15 on a database built in an Excel ® version 2016—categorical variables using frequencies and proportions. Means, medians, standard deviation, ratios, and confidence intervals were obtained for numerical variables. The Shapiro–Wilk test was performed to assess the data distribution using parametric or nonparametric statistics based on the test results. Pearson's $X^2$ and Fischer's exact test were performed to analyze qualitative or categorical variables according to the expected values and Spearman's correlation coefficient in continuous variables. Simple logistic regression was run to calculate the odds ratio (OR) for variables; the final multivariate model was built stepwise with a probability of 0.20 for the independent variables, running a multiple logistic regression analysis. The Hosmer–Lemeshow goodness of fit test evaluated the model's fit. The confidence level was set at $\alpha = 0.05$, and 95% confidence intervals were estimated.

### Results

One hundred and three patients' clinical records and biopsies were examined; 70 corresponded to controls and 33 to cases. The male-to-female ratio was 2.81 (76:27) for the whole

group, 2.88 (52:18) for controls, and 2.66 (24:9) for cases. The variables associated with the age of the patients at the time of diagnosis of the events (ML or CL) did not show a normal distribution. The median age at which patients in the control group were diagnosed with CL was 23.5 years (95% CI:22–27.72). Notably, seven (21.2%) patients from the cases' group did not report a previous cutaneous lesion; the median age at which the remaining 26 cases were diagnosed with CL was 29.24 years (95% CI: 23.45–35.54). In the cases group, the median age at which ML was diagnosed was 38 years (95% CI: 33–45).

In the case group, the received treatment for the CL event was: 12 patients received meglumine antimoniate, one received miltefosine, one received ketoconazole, seven did not receive treatment because they did not have cutaneous leishmaniasis, and 12 patients were not treated with any medication. Of the treated patients in the cases group, 12 showed skin lesion healing, and 2 (14.3%) experienced therapeutic failure; one was treated with meglumine antimoniate, and one was treated with miltefosine. In the control group, 66 patients were treated with meglumine antimoniate, two were treated with ketoconazole, and in two patients, the treatment regimen was not stated in their clinical records. Among the controls, 44 showed skin lesion healing, while 21 (30.1%) patients experienced therapeutic failure; from those, 20 were treated with meglumine antimoniate and 1 with ketoconazole. There was no information regarding the treatment outcome in three patients from the control group treated with meglumine antimoniate. No statistical differences were found between cases and controls in CL therapeutic failure to meglumine antimoniate (p = 0.211).

The comparison of the analyzed variables between cases and controls is presented in Table 1.

In the bivariate analysis presented in Table 1, the variables significantly differing between cases and controls are the completed treatment for the CL event and carrying a parasite infected with LRV1.

In the cases' group, lesions were located exclusively on the nasal mucosa in 75.7% of the patients, lesions involving both nasal and oral mucosa were observed in 15.2% of the patients, and lesions involving nasal, oral, and pharyngeal mucosa in 9.4% of the patients. Following the classification system proposed by Lessa for ML [35], most of the mucosal lesions (73%) were stage III and IV.

From the group of cases, in patients who presented the CL antecedent (n = 26), the median onset for ML was three years after the episode of CL (95% CI: 0.5–14 years). The presence of LRV1 was not correlated with the elapsed time between the diagnoses of CL and ML manifestation (Spearman coefficient = -0.024, p = 0.90). A comparison within the cases group was conducted to determine whether there was an association between the presence of LRV1 and the occurrence of ML when there was no clear history of CL. Four out of seven patients who did not present symptoms of CL were positive for LRV1, contrasting with seven out of twenty-six patients who presented CL antecedent. However, this data distribution shows no significant difference between carrying or not the virus and the absence of CL clinical symptoms before developing ML (p = 0.186).

From the whole group of analyzed samples, *L. (V.) braziliensis* was identified in sixty; *L. (V.) panamensis* in seventeen; *L. (V.) guyanensis* in two, and *L. (V.) panamensis/guyanensis* as complex in one sample. In twenty-three samples, it was not possible to identify the parasite species. *L. (V.) braziliensis* was present in 63.6% (95% CI,45.2–78.7) of the case samples and 55.7.% (95% 43.6–67.1) of the control samples. *L. (V.) panamensis* was identified in 18.1% (95% CI,8.0–36.1) of the case and 15.7% (95% CI,8.7–26.5) of the control samples (S1 and S3 Tables) without significant difference in species distribution between cases and controls (Fig 1C–1E).

LRV1 was detected in seventeen clinical samples, corresponding to a frequency of 16.5% (95% CI, 10.4–25.1) (S1 and S2 Tables). Six samples, 8.5% (95% CI, 3.8–18.1), corresponded

**Table 1. Bivariate association between mucosal leishmaniasis, other relevant variables, and *Leishmania* RNA Virus 1 infection.**

| Variable | Cases | Controls | OR (95% IC) | P-value[a] |
|---|---|---|---|---|
| | n = 33 (%) | n = 70 (%) | | |
| Sex | | | | |
| Male | 24 (72.7) | 52 (74.3) | 0.92 (0.33–2.69) | 0.866 |
| Female | 9(27.3) | 18 (25.7) | | |
| The age group for the CL event* | | | | |
| <15 years | 6(23.1) | 12(17.1) | | |
| 15–44 years | 18(69.2) | 48(68.6) | 0.68 (0.20–2.55) | 0.508 |
| >45 years | 2 (7,7) | 10 (14.3) | 0.5 (0.05–2.62) | 0.385 |
| Geographical origin (Natural Regions) | | | | |
| Orinoco/Amazon | 16(48.5) | 27(40.9) | | |
| Other | 17(51.5) | 39(59.1) | 0.73 (0.31–1.70) | 0.474 |
| Location of the skin lesion for the CL event* | | | | |
| Above the waist | 11(44.0) | 42(60.8) | | |
| Below the waist | 7(28.0) | 13(18.8) | 1.42 (0.82–2.45) | 0.202 |
| Whole body | 7(28.0) | 14(20.2) | | |
| Number of lesions for the CL event* | | | | |
| 1 lesion | 13(52.0) | 42(60.0) | 1.38 (0.49–3.82) | 0.486 |
| >1 lesion | 12(48.0) | 28(40.0) | | |
| Type of treatment for the CL event | | | | |
| Meglumine antimoniate | 8(88.9) | 59(96.7) | 3.68 (0.05–76.5) | 0.685 |
| Other | 1(11.1) | 2(3.3) | | |
| Complete treatment for the CL event | | | | |
| Yes | 9(27.3) | 61(89.7) | 0.043 (0.01–0.14) | 0.0001 |
| No | 24(72.7) | 7(10.3) | | |
| Number of treatment cycles for CL | | | | |
| 1 cycle | 8(88.9) | 40 (65.6) | 0.23 (0.005–2.01) | 0.306 |
| >1 cycle | 1 (11.1) | 21 (34.4) | | |
| Number of CL episodes | | | | |
| 1 episode | 26(100) | 62(91.2) | 0.39 (0.008–3.56) | 0.710 |
| > 1 episode | 0 | 6(8.2) | | |
| Presence of LRV1 | | | | |
| Yes | 11(34.4) | 6(6.1) | 5.33 (1.55–19.42) | 0.0016 |
| No | 22(65.6) | 31(93.9) | | |
| *Leishmania* species | | | | |
| *Leishmania* spp | 4(12.1) | 19(27.1) | | |
| *L. (V.) braziliensis* | 21(63.6) | 39(55.7) | 2.55 (0.76–8.50) | 0.126 |
| No *L. (V.) braziliensis* | 8(24.2) | 12(17.1) | 3.16 (0.78–12.85) | 0.107 |

The p values are determined by Ficher's exact test or Pearson's X2 according to the expected values.

* Seven patients from the cases' group did not report a previous cutaneous lesion; for these variables, the n is 26

[a] α < 0.05

with biopsies from the control group, and eleven, 33.3% (95% CI, 18.8–51.7) corresponded to samples from the cases group. Regarding the occurrence of LRV1 according to *Leishmania* species, twelve LRV1 positive samples corresponded to *L. (V.) braziliensis* (cases:7, controls:5), two samples to *L. (V.) panamensis* (cases:2, controls:0), and three samples to unidentified

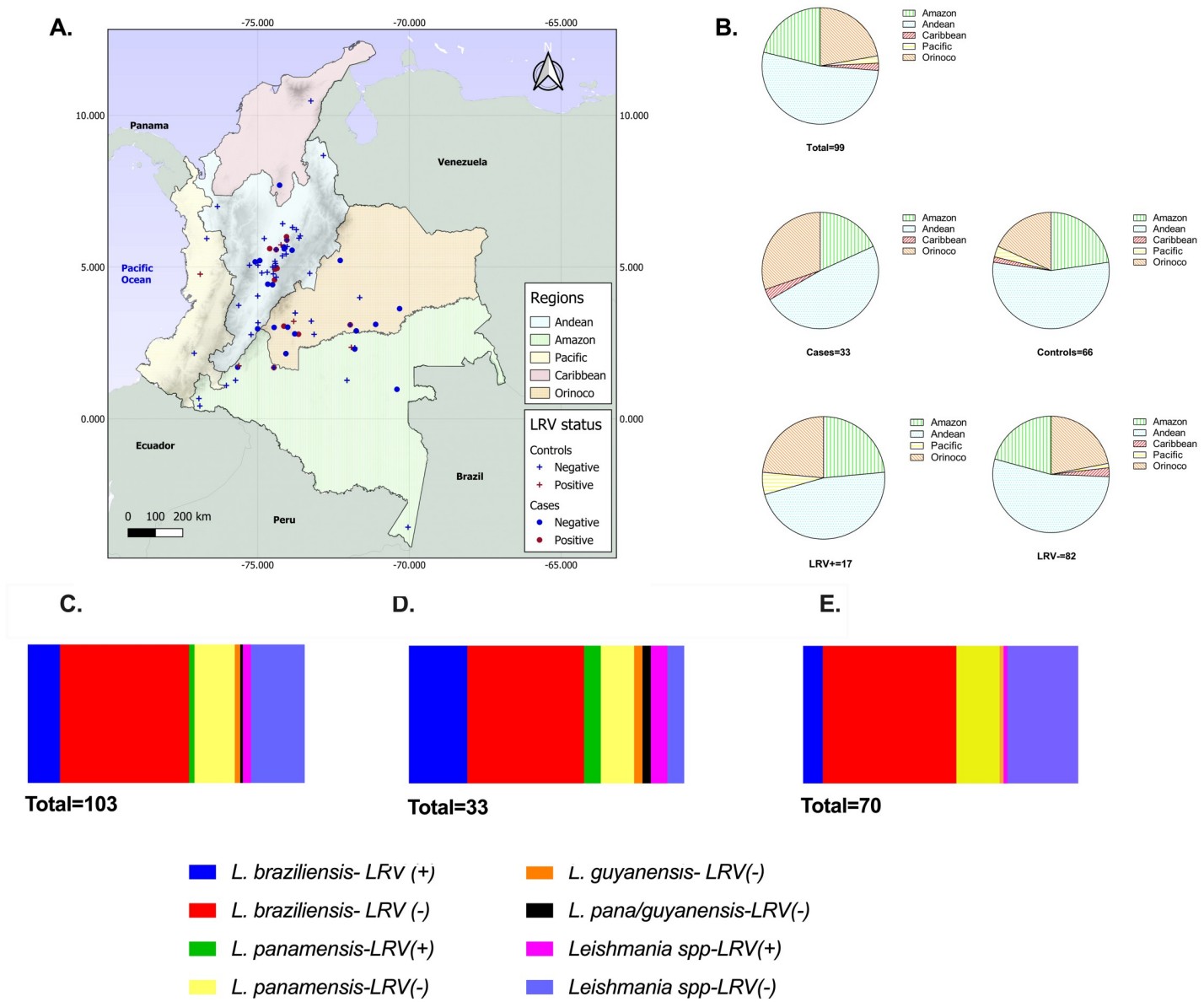

**Fig 1. Distribution of samples according to the geographic origin of the infection, *Leishmania* species, and LRV1 statutes. A**. Presumptive geographical origin of the acquired infection. Cases are represented as dots, and controls as crosses. Red symbols correspond with samples LRV1(+), and blue ones with LRV1(-). **B**. Geographic origin distribution of the total group analyzed (top pie), cases (middle left pie), controls (middle right pie), samples LRV1 (+) (bottom left pie), and samples LRV1(-) (bottom right pie). The number on the bottom of each pie corresponds with the n. Panels **C.- E.** present the distribution of the species identified by hsp70 PCR-RFLP or miniexon PCR in cryopreserved biopsies according to the corresponding patient group and the LRV1 status of the sample. The panel presented in **C.** corresponds to the species distribution and LRV1 status of the whole group, **D.** to the cases group, and **E.** to the control group. Analyses of species distribution between cases/control groups and between LRV1(+) /LRV1(-) were performed using Pearson's X2 test and Fisher's exact test, and no significant differences were found. *L. panamemsis/guyanensis* corresponds to samples that presented an undistinguishable pattern between *L. (V.) guyanensis* and *L. (V.) panamensis*. Shadows on the map correspond with the Andes. The map was created using QGIS 3.38.3 (https://qgis.org/license/). The shapefiles were obtained from publicly available and open-access sources: Natural Earth Data - 10m Physical Vectors (https://www.naturalearthdata.com/downloads/10m-physical-vectors/10m-ocean/), DIVA-GIS Data (https://diva-gis.org/data.html) and GADM (https://gadm.org/download_country.html).

*Leishmania* species (cases:2, controls:1) (S2 Table). There was no significant difference in distribution by species between LRV1 negative and positive samples.

Information regarding the LRV1 status, parasite species, control follow-up, and geographical origin for each sample is presented in S1 Table and Fig 1. Concerning the geographic origin of the infection, most of the evaluated samples came from the Andean region (52.5%), followed by samples from the Orinoco (22.2%) and Amazon regions (21.2%). Pacific and Caribbean regions, as the infection geographic origin, were found only in 4% of the evaluated sample. In 3.8% of the revised clinical records, there was no information regarding the geographical origin of the infection. The LRV1 distribution was found in samples from all those regions except the Caribbean. Evaluation of the LRV1 positivity according to the region where the sample came from shows that 19% of the samples from the Amazon carried the virus, 18.2% of the samples from the Orinoco, and 15.4% from the Andean. Despite the underrepresentation of the Pacific region, one out of two samples of *L.* (*V.*) *braziliensis* was found to be LRV1 infected (S1 Table).

The bivariate analysis for assessing factors that could show differences between case and control groups identified complete treatment and LRV1 infection as factors with statistically significant differences between groups (Table 1). Based on that analysis, variables with $p < 0.20$ were selected to be incorporated into the logistic regression analysis (Table 2).

The bivariate analysis identified two variables with a potentially significant association with the occurrence of ML. These were complete treatment for the CL event (OR = 0.043; 95% CI, 0.01–0.14) and detectable LRV1 (OR = 5.33; 95% CI, 1.55–19.42). The model in the multivariate analysis showed the presence of LRV1 associated with ML (OR = 6.30; 95% CI,1.52–26.10; p = 0.011) and the fact of receiving a complete treatment for the CL event as a protective factor for ML occurrence (OR = 0.039; 95% CI, 0.01–0.12; p = 0.0001) (Table 2).

## Discussion

ML is a disturbing clinical complication associated mainly with infections from New World *Leishmania* species with an uncertain incidence. There is enough evidence supporting the role of the patient's immune system in triggering ML [15, 16]. However, insufficient knowledge is yet to guide clinical decisions or well-defined prevention strategies, such as early progression markers from the CL to the ML form of the disease. This case-control study identified an association between the presence of LRV1 in clinical samples and the occurrence of ML (OR = 6.30; 95% CI,1.52–26.10; p = 0.011), and the multivariate analysis showed that providing a completed treatment for CL event decreased the risk of the presentation of ML [OR, 0.039 (95% IC,0.01–0.12); p,0.0001].

The frequency of LRV1 in the whole group of samples was relatively low at 16.5% compared with other Latin-American reports [24–26, 36]. The rate of LRV1 was higher in cases

**Table 2. Multivariate analysis of factors associated with mucosal leishmaniasis.**

| Variable | OR (95% IC) | $P^a$ |
|---|---|---|
| Complete treatment for the CL event | | |
| Yes | 0.039 (0.01–0.12) | <0.0001 |
| No | | |
| Presence of LRV1 | | |
| Yes | 6.30 (1.52–26.10) | 0.011 |
| No | | |

The final model was built stepwise, and a probability of 0.20 was chosen. The final model included 101 patients.

Hosmer–Lemeshow model adjustment: $X^2 = 0.67$, p = 0.4144

[a] $\alpha < 0.05$

compared to controls, as shown in Table 1. This study also examined the potential link between the parasite species and mucosal leishmaniasis (ML) occurrence. It found that *L. (V.) braziliensis* was the main species in both groups of patients (cases: 21, controls: 39), followed by *L. (V.) panamensis* (cases: 6, controls:11). There was no significant difference in species distribution between cases and controls, as shown in Fig 1 and Table 1. Nevertheless, it is essential to note that most patients were infected with parasites from the *Leishmania* (*Viannia*) subgenus, which are closely related species [37, 38]. There is a limitation given that it was not possible to identify the *Leishmania* species in 23 samples (cases:4, controls:19).

A recent study analyzed 56 *L. (V.) panamensis* isolates from cutaneous lesions in Panama from 2014–2018 and found that 20% were infected with LRV1 [39]. This contrasts with the observation in Colombia, where around 170 isolates of *L. (V.) panamensis* collected between 1983 and 2021 were found to have none infected with LRV1 [40]. It's worth noting that none of the referenced studies involved *L. (V.) panamensis* isolates from ML patients. However, in the analysis presented here, *L. (V.) panamensis* was found in samples from both groups (cases, 18.1%, and controls, 15.7%), and the only two biopsies infected with *L. (V.) panamensis* carrying LRV1, belonged to the ML group. Taken together, these observations, along with the fact that most of the samples in this study came from the Amazon, Orinoco, and central and east Andean Colombian regions (Fig 1), suggest that the differential geographic distribution and frequencies of LRV1 could be indicative of different parasite gene flows associated with each environment [41].

Cantanhêde´s study implied the association between LRV1 and the occurrence of ML in a cohort of 147 patients (CL,109 and ML, 38) [24]; Kariyawasam did not find a link between specific clinical symptoms and the presence of LRV1, but the study did reveal that the relative viral burden of LRV1 was highest in *L. (V.) braziliensis* isolates associated with ML in a group of American tegumentary leishmaniasis patients [26]; Ito found a high frequency of LRV1 in the mucosa of patients suffering from ML [25], while Pereira found a low frequency of LRV1 in a cohort of CL and ML patients, without LRV1(+) samples between the ML group [27]. This inconclusive evidence could be because ML seems to be a multifactorial disease with regional variations. Still, differences in study designs, including varying lengths of patient follow-up, can produce no comparable results. Based on 103 patients (70 with CL and 33 with ML), this multivariate analysis suggests that contracting an infection with *Leishmania* spp.-LRV1(+) is independently linked to ML. This result aligns with Cantanhêde's [24] study, which had a similar patient sample size and was designed to compare CL vs. ML patients. However, the wide confidence interval of the odds ratio (OR) linking LRV1 with ML observed in the present study indicates that a larger sample size is needed to achieve greater precision in this association.

Moreover, the multivariate analysis revealed that completing the treatment for CL significantly reduced the risk of developing ML [OR, 0.039 (95% CI, 0.01–0.12); p < 0.0001]. Although this study did not follow a prospective design to directly assess the impact of completed CL treatment on preventing ML, the results are consistent with previous findings [42, 43]. They support the long-held but unproven assumption that systemic treatment of CL with the appropriate dosage and duration can effectively prevent ML [44, 45]. The systemic versus local treatment of New World CL is a hot topic of discussion. One of the main arguments supporting local therapies is based on reports of viable *Leishmania* parasites in the original cutaneous scars and the intact nasal mucosa of patients after completing systemic treatment with highly toxic medications [46, 47]. Additionally, even in the present study, it has been observed that some patients adequately treated for CL may still develop ML.

Experimental evidence has shown that *Leishmania* spp. carrying LRV1 can lead to complications such as relapse and parasite metastasis [19, 20]. The authors of observational studies

showing LRV1's association with therapeutic failure in CL treatment with antimony and pentamidine have suggested that LRV1-mediated changes in the human host immune response potentially affect the efficacy of drugs [22, 23]. It is well established that the host immune system plays a crucial role in the response to treatments [48–50]. Therefore, the mechanisms involving each of the variables associated with ML occurrence are likely rooted in the background of the host immune system, and their exact contributions to the multifactorial occurrence of ML may be challenging to assess.

One of the main limitations of retrospective studies in identifying risk factors is finding appropriate controls and ensuring proper follow-up. To address this, we selected 70 controls from patients diagnosed with CL up to 2010, guaranteeing a minimum follow-up of 8 years after the skin condition diagnosis. Among these controls, 33 were followed up for a median of 12.8 years (ranging from 0.5 to 22.16 years) to ensure the absence of ML symptoms; 32 controls had a less than six-month follow-up period, and five patients were not followed up after completing their CL treatment. Consequently, one limitation of the study was the inability to follow up with all the patients in the control group.

The findings from the present study must be considered in the geographical context of the disease occurrence. The Eastern Mediterranean Region accounts for 80% of the CL cases reported worldwide; however, the incidence of ML in immunocompetent patients is concentrated in the South American Region [1], where most infections are associated with species belonging to the *Leishmania (Viannia)* subgenus. Consequently, there is an urgent need for a prospective multicentric study that involves LRV1 detection, parasite species identification, and hopefully parasite and viral load estimation in clinical samples from CL patients, with a proper post-treatment follow-up to assess the actual weight of LRV1 as a prognostic marker. This approach would allow a better understanding of the genesis of ML and its geographical determinants. A randomized controlled trial to assess the effectiveness of CL treatment in preventing ML occurrence is not ethically suitable. Therefore, the role of this factor will continue to be a subject of debate until a better understanding of the immunological factors mediating the CL response to treatment is achieved.

## Supporting information

**S1 Fig. Flow chart for the study selection of cases and controls.** The stepwise approach describes the recruitment process and the criteria for including and classifying the patients and samples.
(TIF)

**S2 Fig. Description of the stepwise approach to laboratory processing of the samples.**
(TIF)

**S1 Table. Detailed description of relevant analyzed variables per sample.**
(PDF)

**S2 Table. Detailed description of qPCR, RT-qPCR, and PCR primary data per sample for LRV1 detection following the stepwise approach.**
(PDF)

**S3 Table. Detailed description of qPCR, RT-qPCR, and PCR primary data per sample for *Leishmania* spp. detection and species classification.**
(PDF)

## Acknowledgments

The authors thank the CDFLLA Staff, particularly Dr. Claudia Colorado, Dr. Carolina Camargo, and Dr. Andres Gonzalez, for collaborating in the search for the medical records and specimens in the CDFLLA biobank; Dr. Luis A. Gamboa for helping with clinical examinations; and Mrs. Emma Lopez for contacting and inviting patients from the control group to participate in the study.

## Author Contributions

**Conceptualization:** Fredy A. Pazmiño, Marcela Parra-Muñoz, Carlos H. Saavedra, María C. Echeverry.

**Data curation:** Fredy A. Pazmiño, Marcela Parra-Muñoz, Sandra Muvdi-Arenas, Clemencia Ovalle-Bracho.

**Formal analysis:** Fredy A. Pazmiño, Marcela Parra-Muñoz, Carlos H. Saavedra.

**Funding acquisition:** Marcela Parra-Muñoz, Clemencia Ovalle-Bracho, María C. Echeverry.

**Methodology:** Carlos H. Saavedra, Sandra Muvdi-Arenas, María C. Echeverry.

**Project administration:** María C. Echeverry.

**Supervision:** María C. Echeverry.

**Writing – original draft:** Fredy A. Pazmiño, Marcela Parra-Muñoz, Carlos H. Saavedra, Sandra Muvdi-Arenas, Clemencia Ovalle-Bracho, María C. Echeverry.

**Writing – review & editing:** Fredy A. Pazmiño, Carlos H. Saavedra, María C. Echeverry.

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
