## [Decision Letter · Decision Letter 0]

16 Sep 2024

PONE-D-24-32682Mucosal Leishmaniasis is associated with the Leishmania RNA Virus and inappropriate cutaneous leishmaniasis treatment.PLOS ONE

Dear Dr. Echeverry,

Thank you for submitting your manuscript to PLOS ONE. After careful consideration, we feel that it has merit but does not fully meet PLOS ONE’s publication criteria as it currently stands. Therefore, we invite you to submit a revised version of the manuscript that addresses the points raised during the review process.

We look forward to receiving your revised manuscript.

Kind regards,

Vyacheslav Yurchenko, Ph.D.

Academic Editor

PLOS ONE

Journal Requirements: When submitting your revision, we need you to address these additional requirements. 1. Please ensure that your manuscript meets PLOS ONE's style requirements, including those for file naming. The PLOS ONE style templates can be found at https://journals.plos.org/plosone/s/file?id=wjVg/PLOSOne_formatting_sample_main_body.pdf and https://journals.plos.org/plosone/s/file?id=ba62/PLOSOne_formatting_sample_title_authors_affiliations.pdf 

Additional Editor Comments:

The work of Pazmiño et al. was assessed by three independent Referees, who found it important and well-done. Nevertheless, they all raised several important points that need to be addressed in revision. Please do so and supply a detailed Rebuttal letter addressing all these critiques along with your revised version of the text.

Reviewers' comments:

Reviewer's Responses to Questions

**Comments to the Author**

1. Is the manuscript technically sound, and do the data support the conclusions?

Reviewer #1: Yes

Reviewer #2: Yes

Reviewer #3: Yes

2. Has the statistical analysis been performed appropriately and rigorously? 

Reviewer #1: Yes

Reviewer #2: Yes

Reviewer #3: Yes

3. Have the authors made all data underlying the findings in their manuscript fully available?

Reviewer #1: Yes

Reviewer #2: Yes

Reviewer #3: Yes

4. Is the manuscript presented in an intelligible fashion and written in standard English?

Reviewer #1: Yes

Reviewer #2: Yes

Reviewer #3: Yes

5. Review Comments to the Author

Reviewer #1: This is an interesting manuscript evaluating whether the presence of LRV1 is associated with mucosal manifestations (ML) of American tegumentary leishmaniasis in Colombia.

Specific points:

Abstract - lines 37-38: Please revise this sentence for clarity as follows: The overall frequency of LRV1 was 16.5% (95% CI, 10.4 – 25.12), being higher in samples from cases than controls [33.33% (95% CI, 18.89 – 51.76) vs. 8.57% (95% CI, 3.82 – 18.10)].”

Methods

Line 113: “therapeutic test” – which one? Please specify it.

Line 117: please replace “18S ribosomal gene” with “18S ribosomal RNA gene”

Line 125: where the “parasite” > where the “Leishmania parasite”

Lines 142-144: Does the definition of complete treatment for the CL episode(s) mean that the patient completed the round(s) of treatment with a successful clinical treatment outcome (i.e. cure)? Please clarify this in the text.

Line 143: the duration of the miltefosine treatment needs to be indicated

Line 174: for the multivariate model, please specify that a multiple logistic regression analysis was performed

Results

Regarding the therapeutic schedule for the patients, did the cure rate for the CL episode(s) vary among subjects treated with meglumine antimoniate or the other drugs?

Discussion

Line 262: “there is not sufficient knowledge to guide clinical decisions…” The authors need to specify on which aspects they are referring to here.

Line 274: “are highly related” – do you mean here “phylogenetically close”?

Regarding the documented demonstration of LRV1 presence in L. panamensis, there is a recent study performed in Panama (Gonzalez et al., 2023), which did not find support for a significant association of LRV1 presence in cultured parasite isolates with disease severity in human leishmaniasis. This needs to be discussed in the manuscript.

Line 297: “also due to study design disparities” – I suggest to add here “including varying lengths of patient follow-up”

The discussion section would benefit from a deeper discussion of the findings of this study and the perspectives for future work. For instance:

(related to lines 324-328): It would be interesting to prospectively follow up CL cases from whom LRV-positive parasites were isolated, and establish whether or not the presence of the virus increases the risk of complications. It would be relevant that such a study assesses the LRV load variability among the LRV+ tested biopsies and if there is a higher Leishmania parasite load in lesions which are due to LRV+ parasites.

Based on published work and the findings of present work, the authors need to elaborate on how a complete treatment for a previous CL episode has a protective role in the occurrence of ML.

The wide 95% confidence interval associated with the OR estimate for LRV1-ML (in the multiple logistic regression analysis) in this study means the sample size is small (as acknowledged by the authors in the discussion section). This does not mean the biological effect is not real. The authors should emphasize this further in the discussion by stating that a bigger sample size is required (instead of “may be required”, line 303) to have a more accurate effect size estimate of the association between LRV1 presence and disease severity (ML presentation) in human tegumentary leishmaniasis.

Other points:

Scientific names are written in italics. Please revise this throughout the text, including Tables and Figures.

The standard nomenclature for the Leishmania subgenera is as follows: Leishmania (Viannia) subgenus

Introduction, lines 60-61, “Pentavalent Antimonial” > pentavalent antimonial

Introduction, line 63 & Methods, line 139, “gender” > sex

Introduction, line 63, “immune polymorphism” > immune response gene polymorphism

Methods, line 174: specify that OR means odds ratio

Results, line 250: “protector” > protective

Discussion, line 287: “duplet” > duet or couple

Figure 1 labels: please replace L. pana/guyanensis with L. panamensis /guyanensis

Supplementary Figure S1: 18S gene > 18S rDNA gene

Supplementary Figure S1: box corresponding to the control group definition. “Meets criteria for case definition (Total n=70)” should be replaced with “Meets criteria for control definition (Total n=70)”

Supplementary Table 2:

-Replace “undeterminated” with “undetermined” (meaning absent Ct values)

-Verify the correct name RPL27 (in some instances it was written as RLP27)

Supplementary Table 2 caption:

-replace “in order to associated them” with “in order to associate them”

-replace “subtalbe” with “subtable”

Reference 40 appears together with ref. 39, thus it gets unnoticed.

Reviewer #2: Although the association between the presence of the viral endosymbiont Leishmania RNA Virus 1 and mucosal leishmaniasis has already been reported in a previous study, reinforcing this finding in other regions is highly relevant for understanding the epidemiology of leishmaniasis in different regions of the Americas. Another important finding that the present work brings is the relationship between inappropriate treatment and the progression of the disease with manifestations of mucosal leishmaniasis. A few minor modifications are suggested below to contribute to the improvement of this interesting study.

The Abstract, Introduction, and Discussion are well presented.

Line 58 and 273: Correct "Leishmania Viannia" to "Leishmania (Viannia)"

Line 323: Correct "Leishmania (V.)" to "Leishmania (Viannia)" or "L. (Viannia)"

Methodology / Results

Some steps of the methodology need better explanation:

Frozen biopsies maintained at −80°C were processed to confirm infection by Leishmania spp. and LRV1 detection: Were the biopsy samples frozen at -80°C without any preservative? For the cases, were mucosal lesion biopsy samples used, and for the controls, cutaneous lesion samples?

1. Leishmania spp. Infection was confirmed by either RT qPCR or qPCR of the 18S ribosomal subunit gene of the parasite: Were all 103 samples subjected to both methodologies? If so, what were the results? Why was the parasite load not quantified, as this could allow for an assessment of whether the presence of LRV1 is associated with parasite load and whether this is associated with the mucosal form of leishmaniasis? The reference cited in this section is from the group, but it references the work of van den Bogaart et al. 2013 (doi: 10.1016/j.ijpddr.2013.11.001); this study uses the human β2-microglobulin (β-2M) mRNA gene, so it would be important not only to cite the original references but also to mention whether there were any modifications. It would be important to know if an endogenous control was used and, if so, what the results were. It's also crucial to know if the samples that could not have the Leishmania species identified were positive in these parasite detection assays and just couldn’t be identified, or if they were negative and the endogenous control was positive.

2. Infecting parasite species were identified via PCR amplification of the miniexon gene [33] and PCR-RFLP of the hsp70 gene using as controls MHOM/BR/75/M4147, MHOM/PA/71/LS94, MHOM/BR/75/M2903, MHOM/BZ/82/BEL 21 and MHOM/BR/73/M2269 [34]: Here again, it would be better to cite the original references for PCR for the miniexon and hsp70 genes. In this sense, I ask, and I think it’s important to add the answers to the article, if all samples were subjected to both protocols, or if one protocol was done and only the negative samples or those that couldn’t have the Leishmania species identified were subjected to the other protocol. It's also noteworthy that the cited PCR hsp70 protocol (Garcia et al. 2004) is very good for Leishmania in culture, but it is known not to have good sensitivity for detecting Leishmania directly in clinical samples, likely due to the long PCR fragment amplified. However, there are other approaches that improve the detection and identification capacity of Leishmania spp., which could be used to improve the quality of the results, which are already very good but have a little over 20% of samples without species identification. This identification seems quite relevant since the presence of LRV1 appears to be more frequent in L. guyanensis and L. braziliensis (from Amazonian areas) than in L. panamensis, a species quite common in Colombia. Figure 1 seems to show that among the samples without species identification, a higher number of LRV1+ samples are in the "cases" group than in the "control" group, and the negatives are more frequent in the "control" group. If we consider that LRV1 is not so frequent in L. panamensis and that the mucosal form should be more associated with L. braziliensis, it would be important to identify as many samples as possible.

3. Supp Fig2: The regions cited in Supp Table 1 (Orinoco, Amazon, Pacific, Andean) could be presented: it is interesting to see that the occurrence of LRV1-containing parasites seems to be concentrated in the central region of the country, between the Andean, Orinoco, and Amazon regions. I understand that this is not the focus of the study, but it seems like something interesting to comment on, especially since this is presented later in the discussion.

4. Line 300: Include the mentioned reference regarding the study by Cantanhede et al.

Reviewer #3: Animal studies have shown definitively that the endobiont dsRNA virus LRV1 is associated with increased pathology and metastasis in animal models. While not strictly equivalent, this has prompted many to extrapolate that in humans, the presence of LRV1 may be associated with increased risk of mucocutaneous leishmaniasis. However, various clinical surveys or retrospective studies have yielded variable results, with some showing increased association of ML with LRV1 and some not. The reasons for this are unknown and given the paucity of data and importance of this question for treatment options, additional studies are needed. This work addresses this nicely, with carefully presented analysis and data.

The authors report that in their study site (Colombia) there is a significant elevation of the risk of ML with the presence of LRV1. I was not entirely sure about some of the authors statements which to me imply that the difference is 26% rather than 16% (example from abstract: Results: The overall frequency of LRV1 was 16.5 (95% CI,10.4 – 25.12) higher in samples from cases than controls [ 33.33 % (95% CI,18.89 – 51.76) vs 8.57% % (95% CI, 3.82 – 18.10]. So this should be clarified but as shown in a table, this is statistically significant. This could be more clearly stated and emphasized in the abstract and discussion..

Another important finding is that failure to complete a treatment course against CL resulted in increased ML likelihood. Data supporting This seemingly obvious finding is in fact difficult to find in the literature and deserves more emphasis - as to me it is understated. Besides the health implications, this may be one factor clouding the assessment of ML-LRV1 associations, as in effect some patients with LRV1 and destined for ML are instead counted as 'not associated'. This has been suggested in several papers that could be cited but this is the first evidence to my knowledge.

This finding as well the the reports that exogenous viral infections also can stimulate metastasis in animal models could explain some of the diverse reports for against ML-LRV1 association, and perhaps this could be briefly discussed.

6. PLOS authors have the option to publish the peer review history of their article (what does this mean?). If published, this will include your full peer review and any attached files.

Reviewer #1: No

Reviewer #2: No

Reviewer #3: No

---

## [Author Response · Author response to Decision Letter 0]

22 Oct 2024

Dear reviewers, thanks for evaluating this work. Please find below the detailed response to your observations and note that the lines referred to in this response correspond with the line numbers in the revised manuscript with Track Changes.

Reviewer #1: This is an interesting manuscript evaluating whether the presence of LRV1 is associated with mucosal manifestations (ML) of American tegumentary leishmaniasis in Colombia.

Specific points:

Abstract - lines 37-38: Please revise this sentence for clarity as follows: The overall frequency of LRV1 was 16.5% (95% CI, 10.4 – 25.12), being higher in samples from cases than controls [33.33% (95% CI, 18.89 – 51.76) vs. 8.57% (95% CI, 3.82 – 18.10)].”

The abstract was corrected in lines 37-39, affirming that:

“The frequency of LRV1 in the 103 patients was 16.5% (95% CI,10.4 – 25.12) being higher in samples from cases [33.33 % (95% CI,18.89 – 51.76) than from controls [8.57% (95% CI, 3.82 – 18.10)].”

Methods

Line 113: “therapeutic test” – which one? Please specify it.

In lines 115-116, it was clarified: “therapeutic test (Deﬁned as the improvement of signs and symptoms one month after starting treatment with pentavalent antimonials),”

Line 117: please replace “18S ribosomal gene” with “18S ribosomal RNA gene”

Replacement is done in line 119.

Line 125: where the “parasite” > where the “Leishmania parasite”

Replacement is done in line 128.

Lines 142-144: Does the definition of complete treatment for the CL episode(s) mean that the patient completed the round(s) of treatment with a successful clinical treatment outcome (i.e. cure)? Please clarify this in the text.

Clarification was done by modifying the sentence in lines 145-148 as follows:

“Complete treatment for CL was considered when patients received a least one complete round of any of the following regimes independently from the therapeutic outcome: antimony salts (20 mg/kg/ day) for 20 days, or miltefosine (1.5 to 2.5 mg/kg/ day) for 28 days, or ketoconazole (600 mg/day) for 28 days”

Line 143: the duration of the miltefosine treatment needs to be indicated

Modification is done in line 147.

Line 174: for the multivariate model, please specify that a multiple logistic regression analysis was performed

Modification is done in line 183.

Results

Regarding the therapeutic schedule for the patients, did the cure rate for the CL episode(s) vary among subjects treated with meglumine antimoniate or the other drugs?

Unfortunately, this retrospective design with a reduced “n” for treatment regimens different from antimonials is not appropriate to describe any variation in the therapeutic outcome, and it is not the focus of the study. However, to describe the information recorded about the therapeutic response, a paragraph was inserted between lines 197-209 as follows:

“In the case group, the received treatment for the CL event was: 12 patients received meglumine antimoniate, one received miltefosine, one received ketoconazole, seven did not receive treatment because they did not have cutaneous leishmaniasis, and 12 patients were not treated with any medication. Of the treated patients in the cases group, 12 showed skin lesion healing, and 2 (14.3%) experienced therapeutic failure; one was treated with meglumine antimoniate, and one was treated with miltefosine. In the control group, 66 patients were treated with meglumine antimoniate, two were treated with ketoconazole, and in two patients, the treatment regimen was not stated in their clinical records. Among the controls, 44 showed skin lesion healing, while 21 (30.1%) patients experienced therapeutic failure; from those, 20 were treated with meglumine antimoniate and 1 with ketoconazole. There was no information regarding the treatment outcome in three patients from the control group treated with meglumine antimoniate. No statistical differences were found between cases and controls in CL therapeutic failure to meglumine antimoniate (p=0.211).”

Discussion

Line 262: “there is not sufficient knowledge to guide clinical decisions…” The authors need to specify on which aspects they are referring to here.

A sentence was introduced in lines 302-304 to expand that affirmation. “but there is not sufficient knowledge yet to guide clinical decisions or well-defined prevention strategies, such as early markers of progression from the CL to the ML form of the disease.”

Line 274: “are highly related” – do you mean here “phylogenetically close”?

Yes, that is exactly what we mean. To clarify that, the paragraph was modified in line 319:

“Nevertheless, it is essential to note that most patients were infected with parasites from the Leishmania (Viannia) subgenus, which are closely related species.”

Regarding the documented demonstration of LRV1 presence in L. panamensis, there is a recent study performed in Panama (Gonzalez et al., 2023), which did not find support for a significant association of LRV1 presence in cultured parasite isolates with disease severity in human leishmaniasis. This needs to be discussed in the manuscript.

The reference was included as number 39 and discussed in lines 332-335 as follows:

“A recent study analyzed 56 L. (V.) panamensis isolates from cutaneous lesions in Panama from 2014-2018 and found that 20% were infected with LRV1[39]. This contrasts with the observation in Colombia, where around 170 isolates of L. (V.) panamensis collected between 1983 and 2021 were found to have none infected with LRV1[40]. It's worth noting that none of the referenced studies involved L. (V.) panamensis isolates from ML patients. However, in the analysis presented here, L. (V.) panamensis was found in samples from both groups (cases, 18.1%, and controls, 15.7%), and the only two biopsies infected with L. (V.) panamensis carrying LRV1, belonged to the ML group. Taken together, these observations, along with the fact that most of the samples in this study came from the Amazon, Orinoco, and central and east Andean Colombian regions (Figure 1), suggest that the differential geographic distribution and frequencies of LRV1 could be indicative of different parasite gene flows associated with each environment [41].”

Line 297: “also due to study design disparities” – I suggest to add here “including varying lengths of patient follow-up”

Modification is done in line 358.

The discussion section would benefit from a deeper discussion of the findings of this study and the perspectives for future work. For instance: 

(related to lines 324-328): It would be interesting to prospectively follow up CL cases from whom LRV-positive parasites were isolated, and establish whether or not the presence of the virus increases the risk of complications. It would be relevant that such a study assesses the LRV load variability among the LRV+ tested biopsies and if there is a higher Leishmania parasite load in lesions which are due to LRV+ parasites.

Based on published work and the findings of present work, the authors need to elaborate on how a complete treatment for a previous CL episode has a protective role in the occurrence of ML.

The wide 95% confidence interval associated with the OR estimate for LRV1-ML (in the multiple logistic regression analysis) in this study means the sample size is small (as acknowledged by the authors in the discussion section). This does not mean the biological effect is not real. The authors should emphasize this further in the discussion by stating that a bigger sample size is required (instead of “may be required”, line 303) to have a more accurate effect size estimate of the association between LRV1 presence and disease severity (ML presentation) in human tegumentary leishmaniasis.

The discussion was written again following the referee’s suggestions.

The main changes in the discussion from line 354 are on, 

• The implications of the completed and systemic treatment for the CL event in preventing ML occurrence.

• The perspectives for future studies for assessing LRV1 detection and load calculations as prognostic markers in CL through prospective studies.

• The current study’s limitations.

About the observation in line 303, this was amended in lines 362-364 as follows:

“However, the wide confidence interval of the odds ratio (OR) linking LRV1 with ML observed in the present study indicates that a larger sample size is needed to achieve greater precision in this association.”

Other points: 

Scientific names are written in italics. Please revise this throughout the text, including Tables and Figures.

Modifications were made in the new discussion and the supplementary tables.

The standard nomenclature for the Leishmania subgenera is as follows: Leishmania (Viannia) subgenus

Modification is done on lines 30 and 319.

Introduction, lines 60-61, “Pentavalent Antimonial” > pentavalent antimonial

Modification is done in line 62.

Introduction, line 63 & Methods, line 139, “gender” > sex

Modification is done in line 64.

Introduction, line 63, “immune polymorphism” > immune response gene polymorphism

Modification is done in line 64.

Methods, line 174: specify that OR means odds ratio

Modification is done in line 182.

Results, line 250: “protector” > protective

Modification is done in line 290

Discussion, line 287: “duplet” > duet or couple

The sentence was removed from the manuscript.

Figure 1 labels: please replace L. pana/guyanensis with L. panamensis /guyanensis

Modification is done in line 278 and in the figure.

Supplementary Figure S1: 18S gene > 18S rDNA gene

Supplementary Figure S1: box corresponding to the control group definition. “Meets criteria for case definition (Total n=70)” should be replaced with “Meets criteria for control definition (Total n=70)”

Corrections were made.

Supplementary Table 2:

-Replace “undeterminated” with “undetermined” (meaning absent Ct values)

-Verify the correct name RPL27 (in some instances it was written as RLP27)

Supplementary Table 2 caption:

-replace “in order to associated them” with “to associate them”

-replace “subtalbe” with “subtable”

All corrections were made in Supplementary Table 2.

Reference 40 appears together with ref. 39, thus it gets unnoticed.

The references were amended. 

Reviewer #2.

Line 58 and 273: Correct "Leishmania Viannia" to "Leishmania (Viannia)"

Correction in lines 30, 59, 417

Line 323: Correct "Leishmania (V.)" to "Leishmania (Viannia)" or "L. (Viannia)"

Correction in lines 30, 59, 417

Methodology / Results

Some steps of the methodology need better explanation:

Frozen biopsies maintained at −80°C were processed to confirm infection by Leishmania spp. and LRV1 detection: Were the biopsy samples frozen at -80°C without any preservative? 

Yes. to state that “Dry” was introduced in line 153.

For the cases, were mucosal lesion biopsy samples used, and for the controls, cutaneous lesion samples?

Yes. It was clarified in lines 153-154.

1. Leishmania spp. Infection was confirmed by either RT qPCR or qPCR of the 18S ribosomal subunit gene of the parasite: Were all 103 samples subjected to both methodologies? If so, what were the results? 

In the supplementary figure 1 of the evaluated manuscript, it is presented that the Leishmania spp. 18S ribosomal gene was detected in the cDNA in 45 controls and 19 cases, as well as in the DNA of 25 controls and 14 cases. However, a new supplementary figure (Supplementary Figure 1.B) was introduced, presenting the step-by-step Lab. procedure to clarify it. Cts obtained for the 18S ribosomal subunit gene are presented in the supplementary Table 3.

Why was the parasite load not quantified, as this could allow for an assessment of whether the presence of LRV1 is associated with parasite load and whether this is associated with the mucosal form of leishmaniasis? 

Parasite load was not quantified because this work aimed to determine if the presence of LRV1 was somehow associated with the occurrence of LRV1. Parasite detection was part of the algorithm to assess what samples were suitable for LRV1 detection. Given that we were using fragments of cryopreserved biopsies without adjusting by tissue weight, we could not compare the parasitic load between samples. The referee is right that it will be desirable to adjust the LRV1 detection according to the parasite load in the clinical sample. That procedure must be standardized for future studies where the researchers control tissue sampling. 

The reference cited in this section is from the group, but it references the work of van den Bogaart et al. 2013 (doi: 10.1016/j.ijpddr.2013.11.001); this study uses the human β2-microglobulin (β-2M) mRNA gene, so it would be important not only to cite the original references but also to mention whether there were any modifications.

The referee is right about the primers for Leishmania spp. 18S ribosomal gene are derived by the protocol described by den Bogaart et al. 2013. However, we performed a completely different protocol described in our published paper, so we cite that work. In the work cited, we properly refer to the primers as the ones designed by den Bogaart et al. 2013. In the procedure performed and described in the cited reference, we did not detect the β2-microglobulin (β-2M) nor run the duplex q-RT-PCR.

It would be important to know if an endogenous control was used and, if so, what the results were.

It is not clear what the referee means by endogenous control. If this is regarding the quality of the RNA, the endogenous control used to assess the cDNA quality was the human gene, rpl27, to avoid false negatives associated with long-term storage samples. As stated in the lines 161-163 and the results presented in Supp Table 2. 

“cDNA quality was assessed via RT-qPCR of a human gene, encoding the ribosomal protein L27 (RPL27) [32], in order to avoid false negatives associated with long-term storage samples, following the algorithm presented in supplementary Fig.1. B. “

Additionally, the step-by-step lab procedure description was introduced as supplementary Figure 1. In the Figure caption, it is now stated that the gene encoding for ribosomal protein L27 (RPL27) was used as an endogenous control.

It's also crucial to know if the samples that could not have the Leishmania species identified were positive in these parasite detection assays and just couldn’t be identified or if they were negative and the endogenous control was positive.

Samples negative for Leishmania spp. 18S ribosomal gene in cDNA and DNA and samples negative for the human gene rpl27 in cDNA were excluded from the study, as presented in supplementary figure 1. B.

2. Infecting parasite species were identified via PCR amplification of the miniexon gene [33] and PCR-RFLP of the hsp70 gene using as controls MHOM/BR/75/M4147, MHOM/PA/71/LS94, MHOM/BR/75/M2903, MHOM/BZ/82/BEL 21 and MHOM/BR/73/M2269 [34]: Here again, it would be better to cite the original references for PCR for the miniexon and hsp70 genes. In this sense, I ask, and I think it’s important to add the answers to the article, if all samples were subjected to both protocols, or if one protocol was done and only the negative samples or those that couldn’t have the Leishmania species identified were subjected to the other protocol.

 It's also noteworthy that the cited PCR hsp70 protocol (Garcia et al. 2004) is very good for Leishmania in culture, but it is known not to have good sensitivity for detecting Leishmania directly in clinical samples, likely due to the long PCR fragment amplified. However, there are other approaches that improve the detection and identification capacity of Leishmania spp., which could be used to improve the quality of the results, These approaches are already very good but have a little over 20% of samples without species identification. This identification seems quite relevant since the presence of LRV1 appears to be more frequent in L. guyanensis and L. braziliensis (from Amazonian areas) than in L. panamensis, a species quite common in Colombia. Figure 1 seems to show that among the samples without species identification, a higher number of LRV1+ samples are in the "cases" group than in the "

---

## [Decision Letter · Decision Letter 1]

3 Nov 2024

PONE-D-24-32682R1Mucosal leishmaniasis is associated with the Leishmania RNA Virus and inappropriate cutaneous leishmaniasis treatment.PLOS ONE

Dear Dr. Echeverry,

Thank you for submitting your manuscript to PLOS ONE. After careful consideration, we feel that it has merit but does not fully meet PLOS ONE’s publication criteria as it currently stands. Therefore, we invite you to submit a revised version of the manuscript that addresses the points raised during the review process.

We look forward to receiving your revised manuscript.

Kind regards,

Vyacheslav Yurchenko, Ph.D.

Academic Editor

PLOS ONE

Journal Requirements:

Additional Editor Comments:

Please address the last set of reviewer's comments. I do not envision another round of review provided that all the concerns are adequately addressed in the text and the rebuttal letter.

Reviewers' comments:

Reviewer's Responses to Questions

**Comments to the Author**

1. If the authors have adequately addressed your comments raised in a previous round of review and you feel that this manuscript is now acceptable for publication, you may indicate that here to bypass the “Comments to the Author” section, enter your conflict of interest statement in the “Confidential to Editor” section, and submit your "Accept" recommendation.

Reviewer #1: All comments have been addressed

2. Is the manuscript technically sound, and do the data support the conclusions?

Reviewer #1: Yes

3. Has the statistical analysis been performed appropriately and rigorously? 

Reviewer #1: Yes

4. Have the authors made all data underlying the findings in their manuscript fully available?

Reviewer #1: Yes

5. Is the manuscript presented in an intelligible fashion and written in standard English?

Reviewer #1: Yes

6. Review Comments to the Author

Reviewer #1: In this revised version of the manuscript, the authors have significantly improved the manuscript and adequately addressed my concerns. The paper is ready for publication with a few editions in the text to be incorporated, as pointed out below.

Methods, line 114: regarding the “therapeutic test”. For me the clarification provided by the authors does not correspond to a test of cure; rather, it corresponds to the definition of the therapeutic clinical outcome (i.e. clinical cure) at the time point of follow-up assessment (one month after treatment). This needs to be revised in the manuscript.

Discussion, line 380: please replace “parasitic” with “parasite” (i.e. parasite load estimation)

Scientific names should be written in italics, some instances need to be amended, please revise this thoroughly.

7. PLOS authors have the option to publish the peer review history of their article (what does this mean?). If published, this will include your full peer review and any attached files.

Reviewer #1: No

---

## [Author Response · Author response to Decision Letter 1]

21 Dec 2024

Dear Editor and Reviewers,

Please find below the response to each one of your observations.

Journal Requirements:

Please review your reference list to ensure that it is complete and correct. If you have cited papers that have been retracted, please include the rationale for doing so in the manuscript text, or remove these references and replace them with relevant current references. Any changes to the reference list should be mentioned in the rebuttal letter. If you need to cite a retracted article, indicate the article’s retracted status in the References list and also include a citation and full reference for the retraction notice.

All the references have been reviewed and adjusted according to Plos One instructions. 

Changes were introduced in references 6, 13, 14, 29,30,31, 42 and 43.

Reference number 29 was corrected, given that this reference was amended in 2016. The erratum was incorporated into the citation according to what appears in PubMed Central. The amendment concerned the probe sequence they presented in 2014 for detecting the human B-2 microglobulin gene. However, our work did not involve the detection of that gene. Therefore, to make clear that we use just the primers for the Leishmania spp 18S ribosomal RNA gene published in that paper, but that the method is the one described previously by us in reference number 30, the sentence on lines 120-121 was amended to “to establish the presence of Leishmania spp., via qPCR for the 18S ribosomal RNA gene the primers referred by van den Bogaart E [29] were used following the protocol modified as described [30].” 

Responses to Reviewers

Reviewer #1: In this revised version of the manuscript, the authors have significantly improved the manuscript and adequately addressed my concerns. The paper is ready for publication with a few editions in the text to be incorporated, as pointed out below.

Methods, line 114: regarding the “therapeutic test”. For me the clarification provided by the authors does not correspond to a test of cure; rather, it corresponds to the definition of the therapeutic clinical outcome (i.e. clinical cure) at the time point of follow-up assessment (one month after treatment). This needs to be revised in the manuscript.

The correction was made on lines 114-115, stating that it corresponds with “clinical response to treatment.”

Discussion, line 381: please replace “parasitic” with “parasite” (i.e. parasite load estimation)

The correction was made on line 381

Scientific names should be written in italics; some instances need to be amended, please revise this thoroughly.

Changes were made in Table 1. Lines 498, 502 and 503.

Kind Regards,

Maria C. Echeverry G.

---

## [Editor Report · Decision Letter 2]

23 Dec 2024

Mucosal leishmaniasis is associated with the Leishmania RNA Virus and inappropriate cutaneous leishmaniasis treatment.

PONE-D-24-32682R2

Dear Dr. Echeverry,

We’re pleased to inform you that your manuscript has been judged scientifically suitable for publication and will be formally accepted for publication once it meets all outstanding technical requirements.

Kind regards,

Vyacheslav Yurchenko, Ph.D.

Academic Editor

PLOS ONE

Additional Editor Comments (optional):

I find this paper an important contribution to the field. Kudos to authors!
---

## [Editor Report · Acceptance letter]

10 Jan 2025

PONE-D-24-32682R2 

PLOS ONE

Dear Dr. Echeverry, 

I'm pleased to inform you that your manuscript has been deemed suitable for publication in PLOS ONE. Congratulations! Your manuscript is now being handed over to our production team.

Kind regards, 

on behalf of

Professor Vyacheslav Yurchenko 

Academic Editor

PLOS ONE